# MoleCLUEs: Molecular Conformers Maximally In-Distribution for Predictive Models

**Michael Maser**
Prescient Design, Genentech
South San Francisco, CA
maserm@gene.com

**Nataša Tagasovska**
Prescient Design, Genentech
South San Francisco, CA
natasa.tagasovska@roche.com

**Jae Hyeon Lee**
Prescient Design, Genentech
South San Francisco, CA
leej226@gene.com

**Andrew M. Watkins**
Prescient Design, Genentech
South San Francisco, CA
watkina6@gene.com

## Abstract

Structure-based molecular ML (SBML) models can be highly sensitive to input geometries and give predictions with large variance. We present an approach to mitigate the challenge of selecting conformations for such models by generating conformers that explicitly minimize predictive uncertainty. To achieve this, we compute estimates of aleatoric and epistemic uncertainties that are differentiable w.r.t. latent posteriors. We then iteratively sample new latents in the direction of lower uncertainty by gradient descent. As we train our predictive models jointly with a conformer decoder, the new latent embeddings can be mapped to their corresponding inputs, which we call *MoleCLUEs*, or (molecular) counterfactual latent uncertainty explanations (Antorán et al., 2021). We assess our algorithm for the task of predicting drug properties from 3D structure with maximum confidence. We additionally analyze the structure trajectories obtained from conformer optimizations, which provide insight into the sources of uncertainty in SBML.

## 1 Introduction

Machine learning (ML) approaches have shown great potential to accelerate drug discovery (Jayatunga et al., 2022; Kirkpatrick, 2022), resulting in a plethora of ML-based algorithms preempting traditional molecular design pipelines (Wu et al., 2022; Nori et al., 2022; Stärk et al., 2022). Among the most promising methods are those that leverage 3D structure representations, since molecular function (particularly in biological settings) is directly dependent on atomic structure (Verma et al., 2010; Zheng et al., 2017). Specifically, structure representations are those containing 3D positions of the atoms, atom and bond types, and (optionally) torsion angles.

In live drug discovery programs, access to high-fidelity geometries is often severely limited due to experimental and resource challenges. For example, X-ray measurements, if obtainable, are both expensive and time-consuming and quantum-accurate simulations are prohibitively expensive in most settings (Rackers et al., 2022). With the advancements in 3D molecular-structure prediction (Fu et al., 2022; Somnath et al., 2021; Isert et al., 2023; Askr et al., 2022; Ganea et al., 2021) and less expensive quantum approximations (van der Kamp & Mulholland, 2013; Riniker & Landrum, 2015; Nakata et al., 2020), it is becoming increasingly common to train structure-based ML (SBML) models with *predicted* geometries, resulting in the following procedure for property prediction of, e.g., drug candidates:

NeurIPS 2023 AI for Science Workshop.

- **Step 1.** Obtain 3D structures of training data, either via experiment or computational prediction (generating conformers, Appendix A);
- **Step 2.** Train a property predictor with the conformers obtained in Step 1 (subsection A.1);
- **Step 3.** At inference time, pass computed (predicted) conformers of new candidate(s) through the predictor from Step 2.

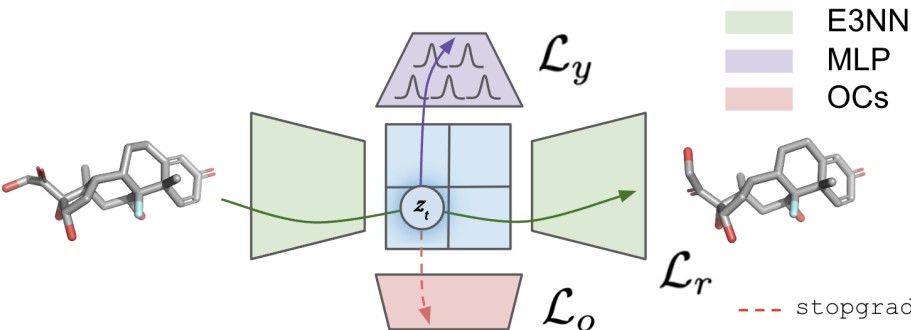

Figure 1: **Step 2. training a structure based predictive model** consisted of a VAE feature extractor - E3NN, property predictor - MLP, and orthonormal certificates - OC, a differentiable uncertainty quantification module.

In practice, Steps 1 and 2 clearly impose problematic biases. Essentially, we assume that new conformers computed in Step 3 will belong to the same distribution of conformers as seen during training, which is difficult to guarantee or even measure. As such, SBML models, e.g., Euclidean neural networks (E3NNs) (Geiger & Smidt, 2022), are prone to poor generalization and often give predictions with very high uncertainties on heldout data (Maser et al., 2023). This is an even greater liability when new samples derive from different structure methods, since each data source may impart structural particularities to datasets, such as preferred bond lengths and angles.

The challenges above are severely problematic for high-risk settings such as ML-based drug discovery (MLDD). The goal of our work is thus to correct or adjust for model biases contributed by 3D structure generation. As a measurable endpoint, we aim at reducing the uncertainty in label predictions for out-of-distribution (OOD) input geometries. We herein present a fully differentiable algorithm to this end called *MoleCLUEs*, which relies on differentiable uncertainty estimators to guide the sampling of learned representations corresponding to novel, in-distribution (ID) conformers.

## 2 Approach & Methods

### 2.1 MoleCLUEs - counterfactual conformers with reduced uncertainty

**Problem setup.** The main task we are concerned with is prediction of molecular properties, either binary classification or regression. We consider a dataset of tuples $\mathcal{D} = (\mathbf{x}, y)_{i=1}^N$ where $\mathbf{x}$ is a canonical representation of a small molecule and $y$ is either binary label or a scalar value of a property of interest such as potency or binding affinity, toxicity, clearance, etc.

**Uncertainty sources in SBML predictive models.** The errors of a SBML model can be contributed to bias, variance, or noise. These three terms relate to either the epistemic (lack of knowledge or data) or aleatoric (inherent data noise) uncertainty Tagasovska & Lopez-Paz (2019). The former motivates the need of confidence intervals that relate to the plausible input space as seen during training, the latter motivates inclusion of predictive variance in results to account for stochasticity in the data. In the context of (3D) molecular property prediction, substantial contributors to both of these uncertainties can be traced to the conformer (i.e., data) generation step (explained in Appendix A) and the predictive model itself (subsection A.1). First, for a given dataset and in general, it is common that some functional groups are better represented than others, which means that when predictors are confronted with a molecule composed of rarer functional groups, confidences (and accuracies) should

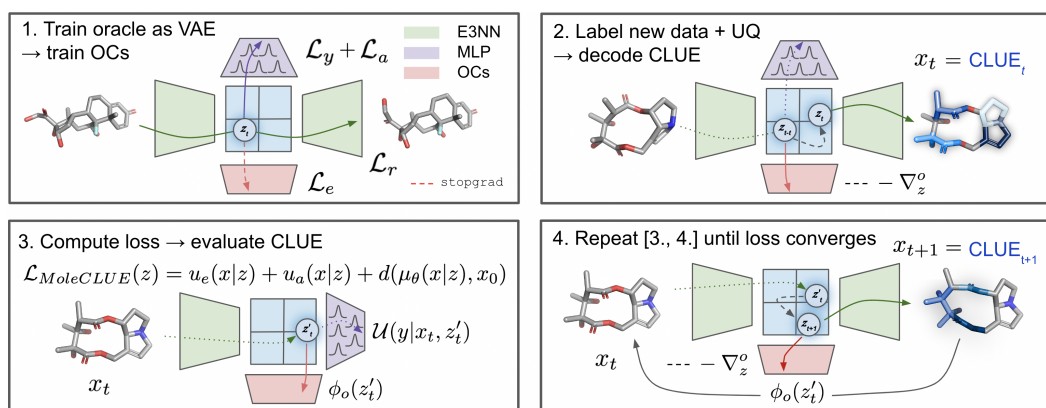

Figure 2: MoleCLUE pipeline for optimizing input conformers with differentiable uncertainty reduction.

naturally be lower. Second, some geometries might simply be too distinct from training conformers, e.g., in shape or asphericity, and hence the model might not be able to provide meaningful prediction for those cases due to lack of support. Both of these difficulties contribute to a higher epistemic uncertainty. Third, heterogeneity in the data might be caused by the different degrees of freedom per molecule that directly influence the number of possible conformers, which imparts noisiness in the predicted conformers for that molecule, e.g., if they're unreasonably diverse. Finally, additional stochasticity might be a result of systematic bias, such as choice of conformer generator method, type of training data (i.e. imprecise X-ray measurements, chemoinformatic tools, human factor). These latter two sources, additionally and critically including label error (especially in wet-lab experiments), further contribute to the aleatoric uncertainty.

We therefore note that for high fidelity models, it is important to address both sources of uncertainties, that is, account and provide estimates for both when leveraging ML models for molecular property prediction. To this end, we include two uncertainty quantification modules in our SBML predictors (1) orthonormal certificates (Tagasovska & Lopez-Paz, 2019) for OOD/epistemic uncertainty, and (2) probabilistic predictions with estimated posterior variance for aleatoric uncertainty. An overview of such SBML predictor is represented in Figure 1. With a structured framework for evaluating uncertainties, we undertake the following challenge:

*Can we improve SBMLs predictive performance for new molecules by reducing the uncertainty stemming from their corresponding predicted conformer(s)?*

In what follows, we provide evidence that we can, and do so by generating *counterfactual* conformers – *MoleCLUEs*, by sampling optimal latent representations in the direction of smaller uncertainty.

**MoleCLUEs** *CLUEs* (Antorán et al., 2021) are based on the idea of counterfactual explanations. We use the term "counterfactual" in the sense of "what would have happened if things had been different?" We adopt a formulation from the interpretability community that uses "counterfactual explanations" as a case of contrastive explanations (Dhurandhar et al., 2018; Byrne, 2019) that assesses how minimal changes in the input reflect over the output predictions. Our MoleCLUEs will thus seek to make small changes to an input conformer in order to reduce the uncertainty assigned to it by our SBML model. To impose that changes are indeed small, we are concerned with counterfactuals $x$ that are close to an original conformer $x_0$, according to some pairwise distance metric $d(x, x_0)$. Having a desired outcome $y_0^c$ in mind that (potentially) differs from the original one $y_0$ produced by the SBML predictor $f$, in our case a probabilistic MLP, counterfactual explanations $x_0^c$ are generated by solving the following optimization problem:

$$x_i^c = \arg\max_x (f(y = y^c|x) - d(x, x_i)) \quad s.t. \quad y_i = y^c. \tag{1}$$

We cannot simply optimize this objective in a high-dimensional input space because it may result in adversarial conformers which are not actionable (Goodfellow et al., 2014). An alternative that lends to high-dimensional data is to leverage deep generative models to ensure explanations are in-distribution. Antorán et al. (2021) suggest that searching for counterfactuals in the lower-dimensional latent space

of an auxiliary generative model avoids the above issues. We denote an auxiliary latent variable $z$ from a deep generative model: $p_\theta(x) = p_\theta(x|z)p(z)dz$. In our setup, this corresponds to the latent representation of the E3NN VAE introduced in Figure 1. We write the predictive means of the E3NN as $\mathbb{E}p_\theta(x|z)(x) = \mu_\theta(x|z)$ and $\mathbb{E}q_\phi(z|x)(z) = \mu_\phi(z|x)$ from the decoder and encoder respectively. With MoleCLUEs, we aim to find points in latent space which decode into conformers similar to our original observation $x_0$ but are assigned low uncertainty by some differentiable estimate of uncertainty $H$, such as those described above.

This goal is achieved by minimizing the following objective:

$$\mathcal{L}(z) = H(y|\mu_\theta(x|z)) + d(\mu_\theta(x|z), x_0). \tag{2}$$

CLUEs are then decoded as:

$$x_{CLUE} = \mu_\theta(x|z_{CLUE}) \quad where \quad z_{CLUE} = \arg\min_z \mathcal{L}(z). \tag{3}$$

The pairwise distance metric takes the form $d(x, x_0) = \lambda_x d_x(x, x_0) + \lambda_y d_y(f(x), f(x_0))$ such that we can enforce a degree of similarity between original data and CLUEs in both input (conformer) and output (predicted property) space. The hyperparameters $\lambda_x$ and $\lambda_y$ control the trade-off between producing low variance CLUEs and CLUEs which are close to the original inputs. In this work, we take $d_x(x, x_0) = ||x - x_0||_2$, which is implemented as `MSELoss` and can be translated to conformer root-mean-squared deviation RMSD $= \sqrt{d_x(x, x_0)}$. Following Antorán et al. (2021), we treat $\lambda_y = 0$ herein, i.e., do not enforce that label predictions must remain similar to those of original data.

We integrate the CLUE module above in an overall pipeline presented in Figure 2. Namely, our differentiable estimate $H$ consists of two terms: (1) epistemic or model uncertainty ($u_e$) via orthonormal certificates (OCs, $C$)(Tagasovska & Lopez-Paz, 2019), multiple linear classifiers trained on top of lower dimensional feature representation $\phi(z|x)$, $u_e(x) = ||C \cdot \mu_\phi(z|x)||_2$ (OCs evaluate close to 0 if a conformer's latent is in distribution and far otherwise); and (2) $u_a(y|x) = \sigma(f(\mu_\theta(x|z)))$, or the variance in the posterior over the predictions of the MLP $f$. Having a separate estimate for the different sources in the predictive uncertainty is desirable, as it lets us explore different configurations when optimizing conformers, focusing on reducing the epistemic or the aleatoric portion. The choice can be made based on the starting molecule; for example, a domain expert can determine if the molecule has been sufficiently well represented in the training data, if it is highly unusual, or if its structural degrees of freedom might lead to unrealistic poses. Finally, our MoleCLUE objective loss has the following form:

$$\mathcal{L}_{MoleCLUE}(z) = u_e(x|z) + u_a(x|z) + d(\mu_\theta(x|z), x_0). \tag{4}$$

We note that the MoleCLUEs framework is modular; we can use any differentiable uncertainty estimate, whichever we find most suitable at inference. Further, additional objectives can optionally be added that regularize or enforce desirable latent properties of the newly encoded CLUE within our representation module $\Phi$, such as L2-norm and Kullback-Leibler divergence (see subsection B.1).

## 2.2 E3NNVAE

To enable the generation of MoleCLUEs, we require a decoder module $\Theta$ that takes latent vectors $z$ as input and outputs position matrices $x$, i.e., $\mu_\theta(x|z)$. In particular, we require that:

1. decoded outputs maintain the initial graph structure (i.e., nodes $v_0$ and edges $e_0$) of input molecule $\mathbf{x}_0 = \{v_0, e_0, x_0\}$; and

2. the latent space $\mathcal{Z}$ from which $z$ are sampled must maintain the equivariance of the encoder E3NN (Geiger & Smidt, 2022).

Constructing our generative model as a typical VAE makes achieving 1) a challenge in that global pooling after the encoder module $\Phi$ ablates graph structure, which existing decoder methods may fail to recover (Ganea et al., 2021; Liu et al., 2023; Xu et al., 2023). We therefore implement a novel architecture (depicted in Figure 3) that transfers the hidden node representations $h_\phi$ to the decoder by skip connection. In order to update $h_\phi \to h_\theta$, we treat latent vectors $z$ as "supernodes" and add virtual edges between sampled $z$ and all existing nodes $h_\phi \leftarrow v_0 \in \mathbf{x}_0$. Information is then propagated via a relational graph convolutional network (RGCN, Schlichtkrull et al. (2017)) as $\Theta$,

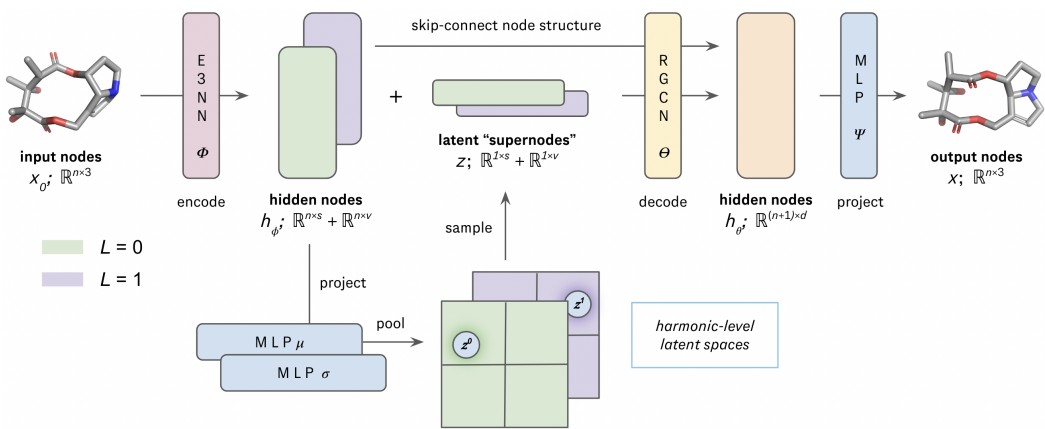

Figure 3: E3NN VAE architecture. Latent distributions are separated by spherical harmonic level $L$ to maintain equivariance in sampling & decoding. $n$ = number of nodes (heavy atoms), $s$ = scalar-feature ($L = 0$) dimensionality (128 herein), $v$ = vector-feature ($L = 1$) dimensionality ($64 \times 3 = 192$ herein), $d$ = hidden-node dimensionality (128 herein).

giving predicted position matrix $x$ after projection of $h_\theta$ with MLP $\Psi$, keeping topology intact (see subsection B.1).

For requirement 2), we construct separate latent distributions $q_\phi^L(z^L|x)$ for each spherical-harmonic level $L$ modeled in $\Phi$ (Geiger & Smidt, 2022). During sampling, we draw from each subspace $q_\phi^L$ independently, and concatenate the reshaped samples to give $z$ that maintains the equivariant tensor structure of $h_\phi$. Details for the remaining E3NNVAE (and OC) components (implementation, training, and hyperparameters) are delegated to Appendix B.

It is worth noting that requirement 1) above can be loosened in settings where decoding an entirely new molecule (2D and 3D graph) is desirable. In this case, we do not require that the 2D topology $\{v, e\}$ of output $\mathbf{x}$ match that of $\mathbf{x}_0$, and thus we can freely decode, e.g., via autoregressive or diffusion-based methods (Ganea et al., 2021; Xu et al., 2023). We leave investigations to this end for future works.

We also note that our VAE construction differs substantially from that in the seminal CLUEs work (Antorán et al., 2021). In Antorán et al. (2021), the CLUE VAE is trained as an auxiliary generative model with a separate "true" data source. Herein, we do not assume access to such additional data, and instead train both our predictor ('oracle') and generator (VAE) jointly and end-to-end as a single network. Beyond representing a novel approach to CLUE modeling, our design directly impacts the optimization stage, in that each loss term in $\mathcal{L}_{(Mole)CLUE}$ is dependent on modules trained jointly under our framework, as opposed to separately in prior art. Future works will seek to understand the implications and differences of using each protocol. Moreover, other applications of counterfactual generation relying on 3D representations could benefit from our modification, e.g., data described as point clouds or mesh grids in graphics, engineering, and biomedical imaging (Rasal et al., 2022b,a; Pawlowski et al., 2020).

## 3 Experiments

### 3.1 Data

In this work, we consider a regression task from a public benchmark dataset, the Therapeutic Data Commons (TDC, Huang et al. (2022), (link)). The task (**Clearance_Hepatocyte_AZ**) is to predict an input drug molecule's rate of clearance from the human body, a critical property for late-stage drug optimization (Di et al., 2012). 3D conformers of all molecules within are computed and processed as in Axelrod & Gomez-Bombarelli (2020); Maser et al. (2023), and we direct the reader to these works for further details.

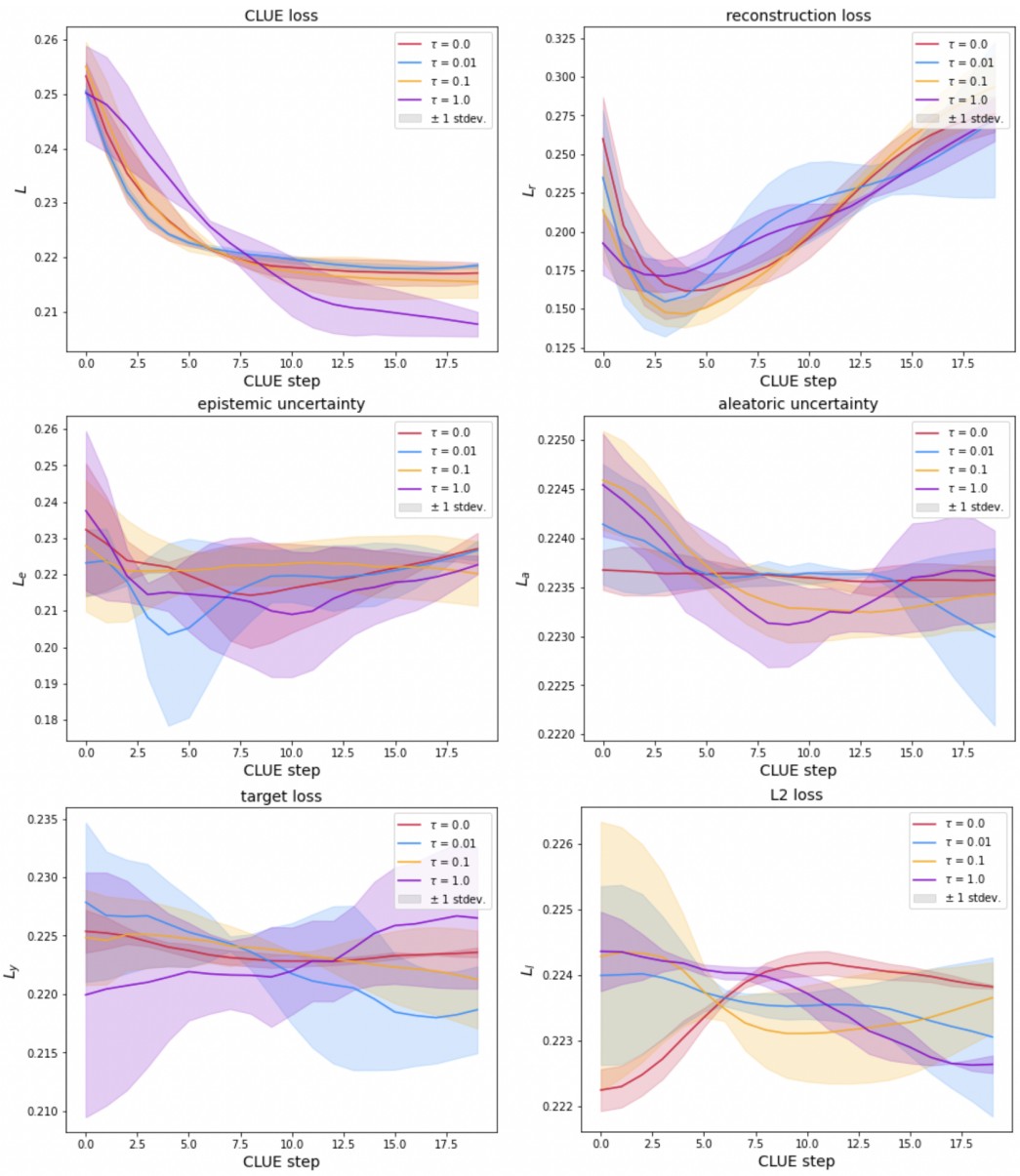

Figure 4: CLUE-optimization loss curves (clue learning rate $= 0.1$). Note that y-axis values are normalized across $\tau$ to be able to visualize important trends.

## 3.2 Experiment setup

To evaluate MoleCLUEs, we use a held-out test set (as given in the TDC), the conformers of which we progressively contaminate with Gaussian noise as in Maser et al. (2023). First, we pass these conformers through our predictive model and rank the results by descending prediction errors and uncertainties, i.e., each term in $\mathcal{L}_{MoleCLUE}$. Then, we select the top-10% of the most difficult molecules by each term and try to improve the predictions by bringing their 3D representation closer to the training data, i.e., optimizing $\mathcal{L}_{MoleCLUE}$. We include results for increasing amounts of noise contamination $\tau \in \{0.0, 0.01, 0.1, 1.0\}$, where $\tau = 0$ (i.e., no noise) corresponds to the original held-out molecules with baseline error & uncertainty. With these different configurations of the analysis, we get to explore the sensitivity and performance improvement of MoleCLUEs.

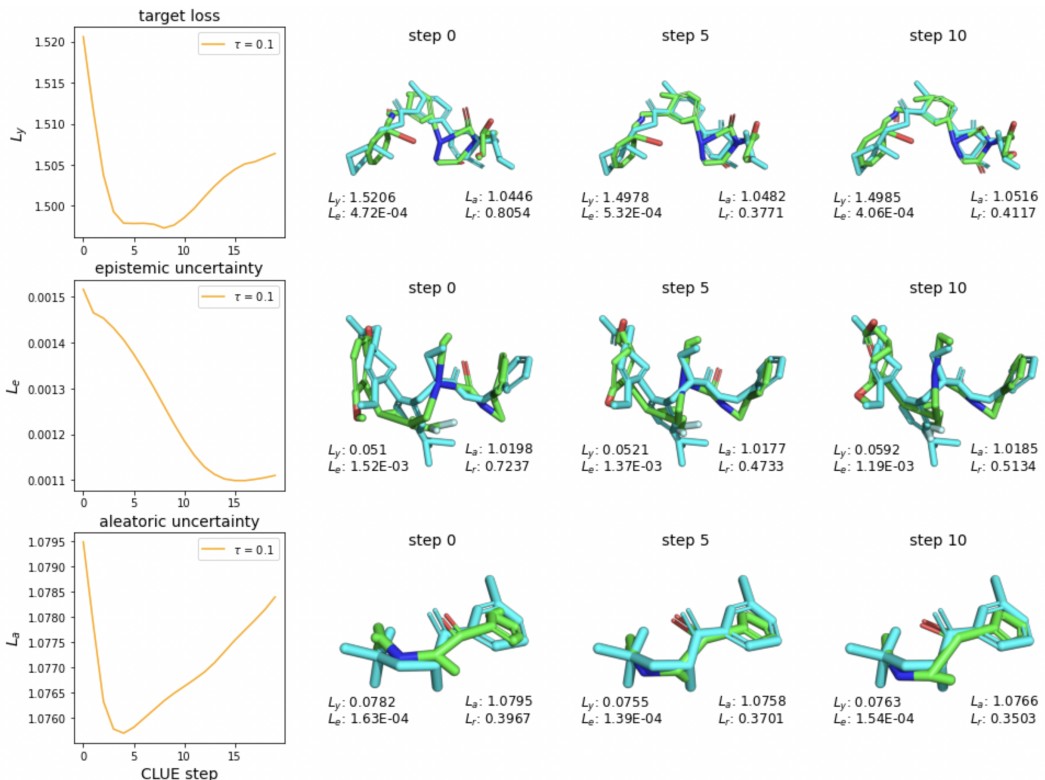

Figure 5: Example CLUE simulations and corresponding loss trajectories; randomly chosen from worst 10% test set examples in respective loss term with original noisy input (cyan). Top) $\mathcal{L}_y$, middle) $\mathcal{L}_e$, bottom) $\mathcal{L}_a$.

## 4 Results & Discussion

### 4.1 Experimental results

Results are shown in Figure 4, aggregated over three runs of the full pipeline shown in Figure 2 and described in subsection 3.2. As can be seen, our method is able to afford substantial reductions in uncertainty on average over CLUE steps, as well as moderate improvements in target prediction error. This effect, as expected, is highly dependent on the level of noise $\tau$ added to the initial conformers. In most cases, intermediate noise levels ($10^{-2} \leq \tau \leq 10^{-1}$) showed strongest and smoothest improvements (blue and yellow curves). On the other hand, uncorrupted examples ($\tau = 0.0$, red curves) showed steady but only minor uncertainty reductions.

With *severe* corruption ($\tau = 1.0$, purple curves), however, strong remediation of both aleatoric and epistemic uncertainties was observed, though the corresponding improvement in target prediction was highly variable. That said, for this noise scale losses had typically not converged after the default 20 iterations; it is yet unclear if additional improvements could be made with longer simulations. Using higher CLUE learning rates (LR), however, we do observe strong convergence in epistemic and aleatoric uncertainty $\mathcal{L}_{e,a}$ as well as target predicion error, (Figure 10), indicating the potential of our method to remedy even severely OOD samples. It is worth noting that early steps were typically accompanied by moves toward the clean, un-noised conformer input (reconstruction loss panel) but tended to diverge in later iterations. Full results are included in subsection C.1, including for sweeps over CLUE LR and normalization hyperparameters.

### 4.2 Optimization trajectories

Individual examples were chosen at random from the experiments above to visualize the structural simulations resulting from our protocol (Figure 5). In each example, as desired, intermediate

conformers are discovered that reduce the problematic loss term while remaining close to the input conformer. Additional randomly selected examples are provided in subsection C.2, which include challenging cases where the loss term of interest was not able to be improved. In particular, examples of initial noisy conformers with high RMSD to the original input ($\mathcal{L}_r = \sqrt{d_x(x, x_0)}$) were challenging to optimize, at least with the hyperparameters studied ($\tau = $ CLUE LR $= 0.1$). Interestingly, and not unexpectedly, we observe in many cases that CLUE conformers continue to diverge from $x_0$ as loss terms other than $d_x(x, x_0)$ are optimized.

It is additionally worth noting that many of the analyzed CLUE conformers contain non-physical sub-graph geometries (distorted bond lengths, angles, etc.). This is also not unexpected, in that we do not include any energetic constraints/evaluations in conformer generation. That said, in line with Figure 4 (top) we do see in all cases that the RMSD to the input conformer (denoted as $L_r$ in each panel of Figure 5) does decrease over the initial CLUE steps. This indicates that the latent vectors we sample in the direction of lower uncertainty also correspond to decoded conformers that are more physically reasonable (as imposed by the distance term in $\mathcal{L}_{MoleCLUE}(z)$), which we consider a desirable result. Future work will investigate the use of physical priors and/or calls to physics-based score functions during CLUE optimization.

## 5 Conclusion

Herein, we presented a novel algorithm *MoleCLUEs* for obtaining molecular conformers that minimize uncertainty and label error in a 3D predictive model, leveraging differentiable uncertainty quantifiers and a novel equivariant conformer generative model. An open question is how our method will perform in inference settings where ground-truth labels $y$ are unavailable, as reducing uncertainty alone may not necessarily reduce label error (i.e., increase accuracy). One avenue for exploration includes training an auxiliary supervised oracle that infers our model's target-prediction error $\mathcal{L}_y$ on heldout data, and including the reduction of this output in $\mathcal{L}_{MoleCLUE}$.

A positive byproduct of our protocol is the inherent interpretability brought on by MoleCLUEs themselves being real conformers (see Figure 5). A practitioner can easily superimpose a starting conformer with higher uncertainty and a MoleCLUE. The difference between the two can pinpoint which parts of the conformers contributed mostly to the uncertainty in the prediction, analogous to Antorán et al. (2021). Other valuable application areas we anticipate include large-molecule (protein, antibody, etc.) property prediction as well as molecular structure prediction and even energetic optimization. Studies toward these ends are ongoing, which we expect to be highly insightful in SBML and MLDD more generally.

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

# A Conformer generation - 3D molecular-structure prediction

Each string representation of a small molecule, corresponds to many conformations, as illustrated in Figure 6a. This is because small-molecules are flexible structures that adopt to multiple *conformations* depending on the solution/room temperature. The conformational space may be very large, and is a function of number of rotatable bonds. Each of those conformers has a different impact of certain drug property such as binding (i.e. the strength that keeps a the drug - small molecule attached to a ligand) or permeability. Figure 6b exemplifies this further, two conformers of the same molecule fit the binding pocket differently.

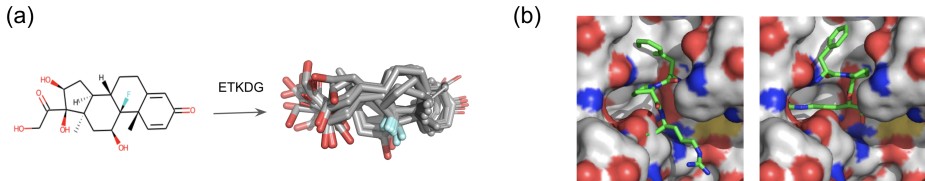

Figure 6: **Step 1. Conformer prediction and its importance.** a) generating conformers with chemo-informatics tools. ETKDG = Experimental Torsion Knowledge Distance Geometry (Riniker & Landrum, 2015). b) same molecule but different conformer implies very different binding (Source: (Ebejer, 2012)).

There are two general approaches for predicting 3D structures (Riniker & Landrum, 2015): (1) a systematic approach: change torsion angles of all rotatable bonds by a small amount, which for large molecules is infeasible; and (2) a stochastic approach: use of random algorithms such as distance geometry, Monte Carlo simulation and genetic algorithms to permute torsion angles. Both approaches make use of statistically derived data from PDB and CSD to determine most common angles between different atom types.

## A.1 Structure-Based Predictive Models

Many recent works have demonstrated molecular property prediction with SB-ML. Approaches of interest here can be broadly categorized into geometric deep learning (GDL) with, e.g., E3NNs Geiger & Smidt (2022), and volumetric deep learning with, e.g., voxel-based models of atomic densityBrock et al. (2016); Sunseri (2021). E3NNs and their derivativesThomas et al. (2018); Batatia et al. (2022); Liao & Smidt (2022) have demonstrated state-of-the-art performance for QM-property prediction from QM/MM-optimal geometries. In these settings, data is often large in scale and highly standardized in geometryPinheiro et al. (2020); Dwivedi et al. (2022). It was recently demonstrated that E3NNs supervised with biochemical assay data can be highly non-smooth under minor perturbations to input geometriesMaser et al. (2023). Volumetric models have been employed for large- and small-molecule property prediction as well as generationSunseri (2021); Guo & Chen (2022). Little is known about their generalizability properties under perturbation and/or distribution shift.

# B Model details

## B.1 E3NNVAE

Next, we divulge in the most significant implementation details of E3NNVAE, while subsection 2.2 describes its development and high-level architecture. Hyperparameters for the E3NN encoder ($\Phi$) were used exactly as in Maser et al. (2023). To render $\Phi$ variational, an additional projection MLP was added to give the latent covariance $\sigma_\phi(z|x)$, with identical structure as the readout MLP in Maser et al. (2023), which gives E3NN mean $\mu_\phi(z|x)$.

As described in subsection 2.2, the latent distributions $q_\phi(z|x) = \mu_\phi(z|x), \sigma_\phi(z|x)$ are sliced into their spherical-harmonic-level components (Geiger & Smidt, 2022) to maintain equivariance in sampling. Up to $L = 1$, this amounts to isolating scalar- and vector-feature distributions $q_\phi^0(z|x)$

and $q_\phi^1(z|x)$, respectively. Sampling from each $q_\phi^L$ independently gives $z^0 \in \mathbb{R}^{128}$ and $z^0 \in \mathbb{R}^{64 \times 3}$, which are flattened and concatenated to give the final latent vector $z \in \mathbb{R}^{320}$.

For target prediction, $z$ is passed to a probabilistic MLP as in Maser et al. (2023). For conformer (CLUE) generation, $z$ is first mixed with penultimate node representations $h_\phi$ from $\Phi$. To achieve this, each $z$ vector in a mini-batch is added as a new "supernode" to its corresponding input graph $\mathbf{x}_0$ and connected to all other nodes $v$ in the graph via a 'virtual' edge-type. Since there are no 3D coordinates associated with supernode $z$, we mix node and edge features of $z$ with all $h_\phi$ via a 2D RGCN module $\Theta$ as described in subsection 2.2 (Schlichtkrull et al., 2017). After initial mixing, $\Theta$ contains four hidden layers of dimension 128, each of which are followed by `ShiftedSoftPlus` activation and `LayerNorm`. The penultimate decoded node representations $h_\theta$ are projected with MLP $\Psi$ to output the predicted position matrix $x'$, which is transformed with the normalization constants of the input position matrix $x_0$ to give the final conformation $x$ after removal of the remnant transformed "supernodes" $z_\theta$.

The overall loss for training our E3NNVAE includes terms for target prediction ($\mathcal{L}_y$), VAE reconstruction ($\mathcal{L}_v$), Kullback-Leibler (KL) divergence ($\mathcal{L}_k$), and latent L2-norm loss for regularization ($\mathcal{L}_l$). Each term is weighed by a $\lambda_i$ hyperparameter, all of which we hold at 1.0 herein. The final formulation is thus $\mathcal{L}_{E3NNVAE} = \frac{1}{N} \sum_{i=1}^{N} \lambda_y \mathcal{L}_y + \lambda_v \mathcal{L}_v + \lambda_k \mathcal{L}_k + \lambda_l \mathcal{L}_l$, where $N$ is the size of the dataset.

### B.2 Orthonormal certificates (OCs)

OCs were implemented exactly as in Tagasovska & Lopez-Paz (2019). 100 certificates (linear classifiers) were used. The orthonormality penalty was weighed equally with training-example recognition (`MSELoss`) during training, i.e., $\lambda_c = 1.0$.

## C Extended Results

### C.1 CLUE optimization curves

Full results including all possible loss terms in $\mathcal{L}_{MoleCLUE}$ are below at log-decreasing CLUE learning rates. In each panel (as in Figure 4), the worst-10% of examples from the test set ranked by the panel's labeled loss term were used for analysis in order to observe optimization effects. Results are aggregated over these examples and show mean and standard deviation over three repeats of the full pipeline experiment, from E3NNVAE and OC training through CLUE optimization. Within each panel and before aggregation, results for each CLUE run at each noise scale $\tau$ are normalized in order to cleanly visualize the relative loss trends between $\tau$'s. Without this, as expected, the raw loss values often exist on drastically different scales, which can obscure the curve slopes at certain $\tau$.

We additionally include results both with and without normalization of the loss terms *during* training of the CLUE optimizer. We note that there are substantial differences in outcome, particularly in reconstruction ($\mathcal{L}_r = d_x(x, x_0)$). Without normalization, this term appears to dominate the overall loss ($\mathcal{L}_{MoleCLUE}$), a reasonable result due to its scale relative to the other terms ($\sim 10^{0\text{-}1}$ vs. $\sim 10^{-4\text{-}-1}$). However, we see that, when normalized, other terms are allowed to overtake $\mathcal{L}_r$ in optimization and in most cases $d_x(x, x_0)$ actually *increases* over CLUE steps. This result may or may not be acceptable depending on the application and on the extent of the conformer divergence with increasing $\mathcal{L}_r$. We leave these choices to the practitioner and note that additional control of term importance can be taken using their weight hyperparameters $\lambda_i$, which we hold at 1.0 for all terms and experiments herein.

### C.1.1 Un-normalized $\mathcal{L}_{MoleCLUE}$ terms

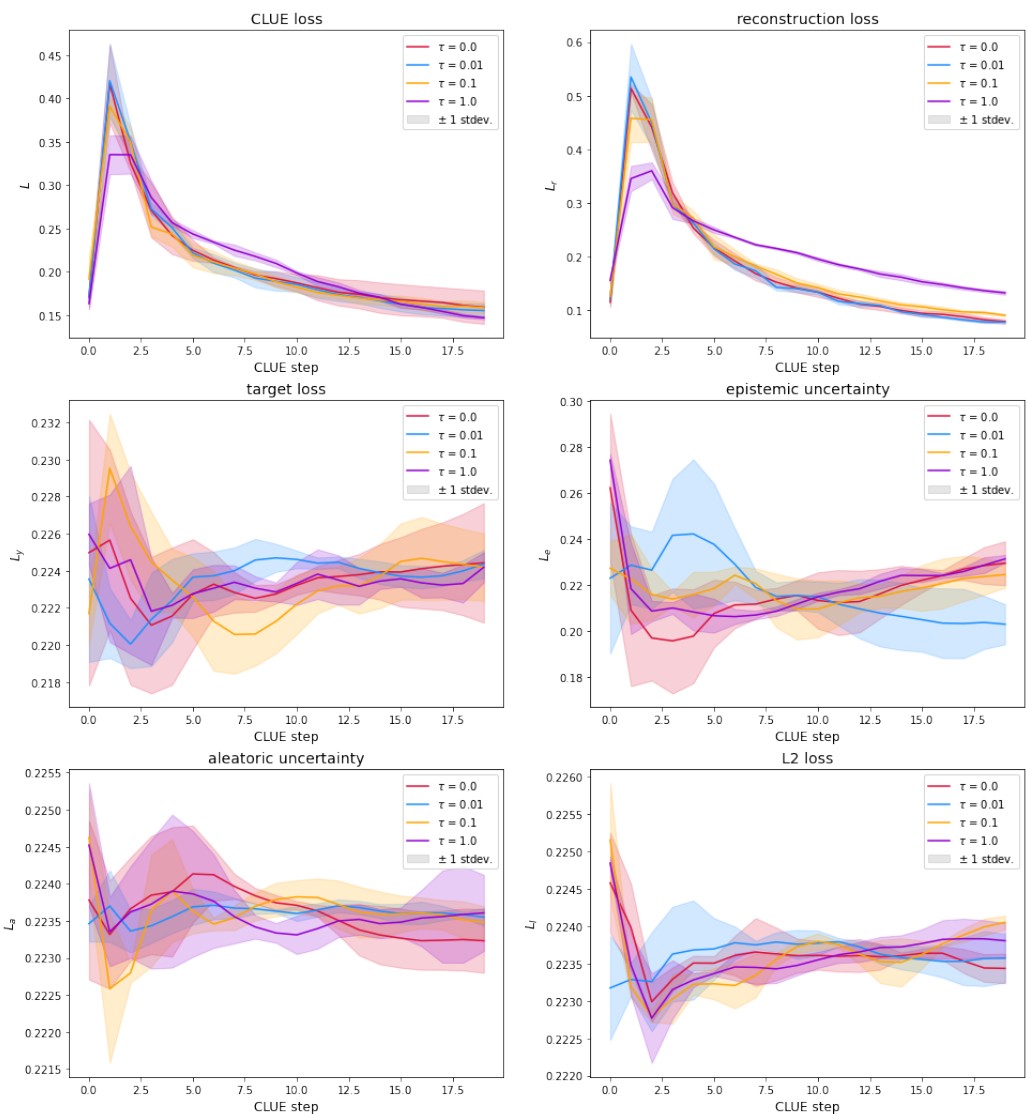

Figure 7: CLUE-optimization loss curves (learning rate $= 1.0$, un-normalized $\mathcal{L}_{MoleCLUE}$ terms).

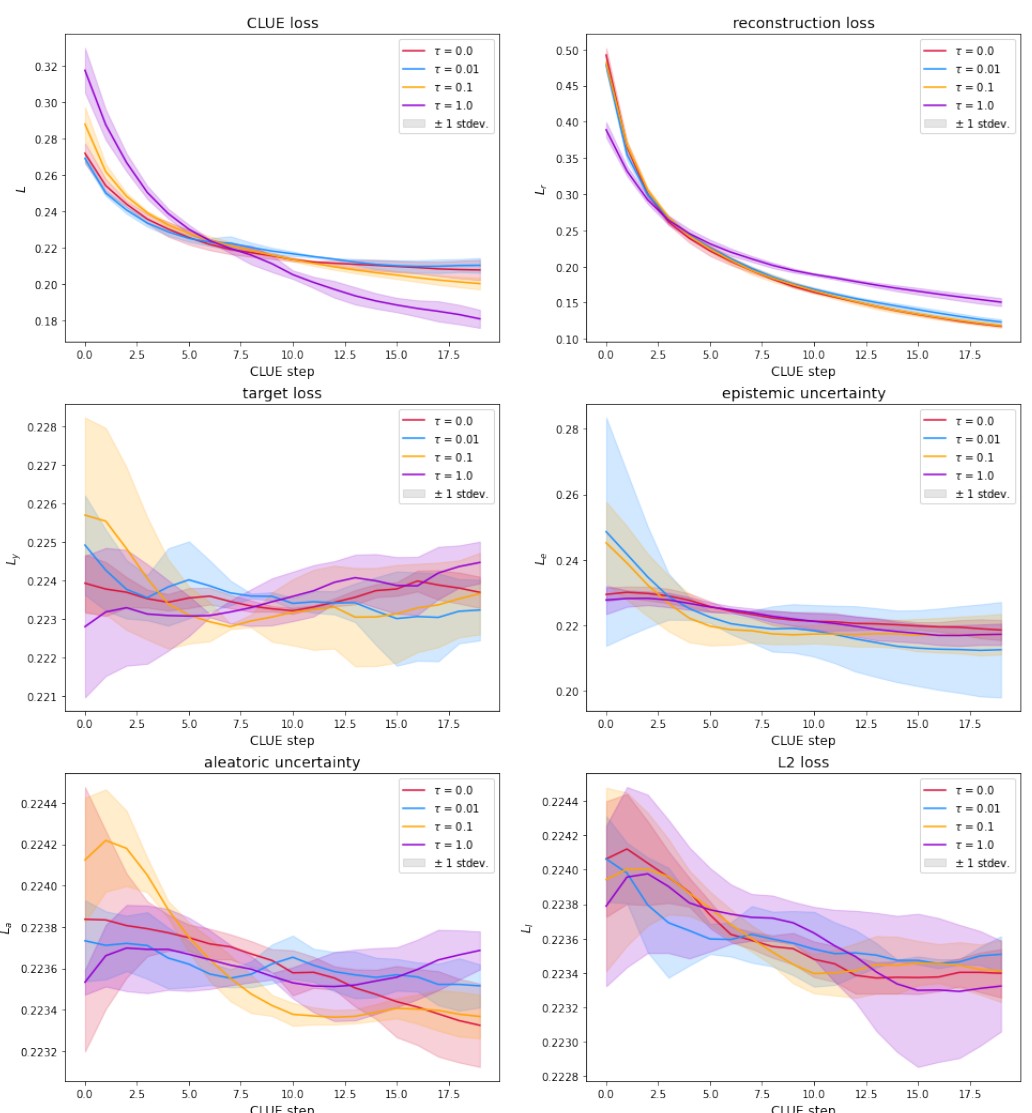

Figure 8: CLUE-optimization loss curves (learning rate $= 0.1$, un-normalized $\mathcal{L}_{MoleCLUE}$ terms).

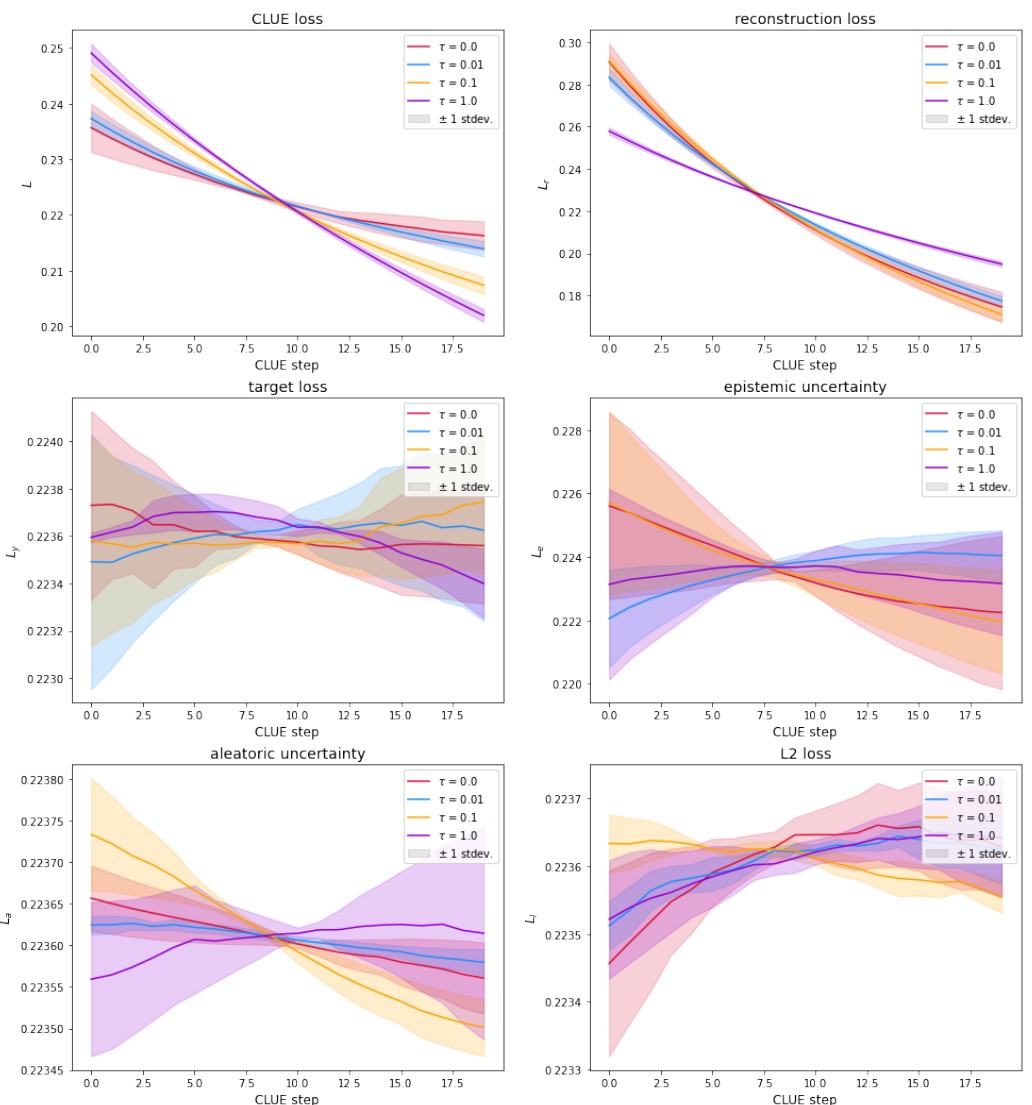

Figure 9: CLUE-optimization loss curves (learning rate $= 0.01$, un-normalized $\mathcal{L}_{MoleCLUE}$ terms).

## C.1.2 Normalized $\mathcal{L}_{MoleCLUE}$ terms

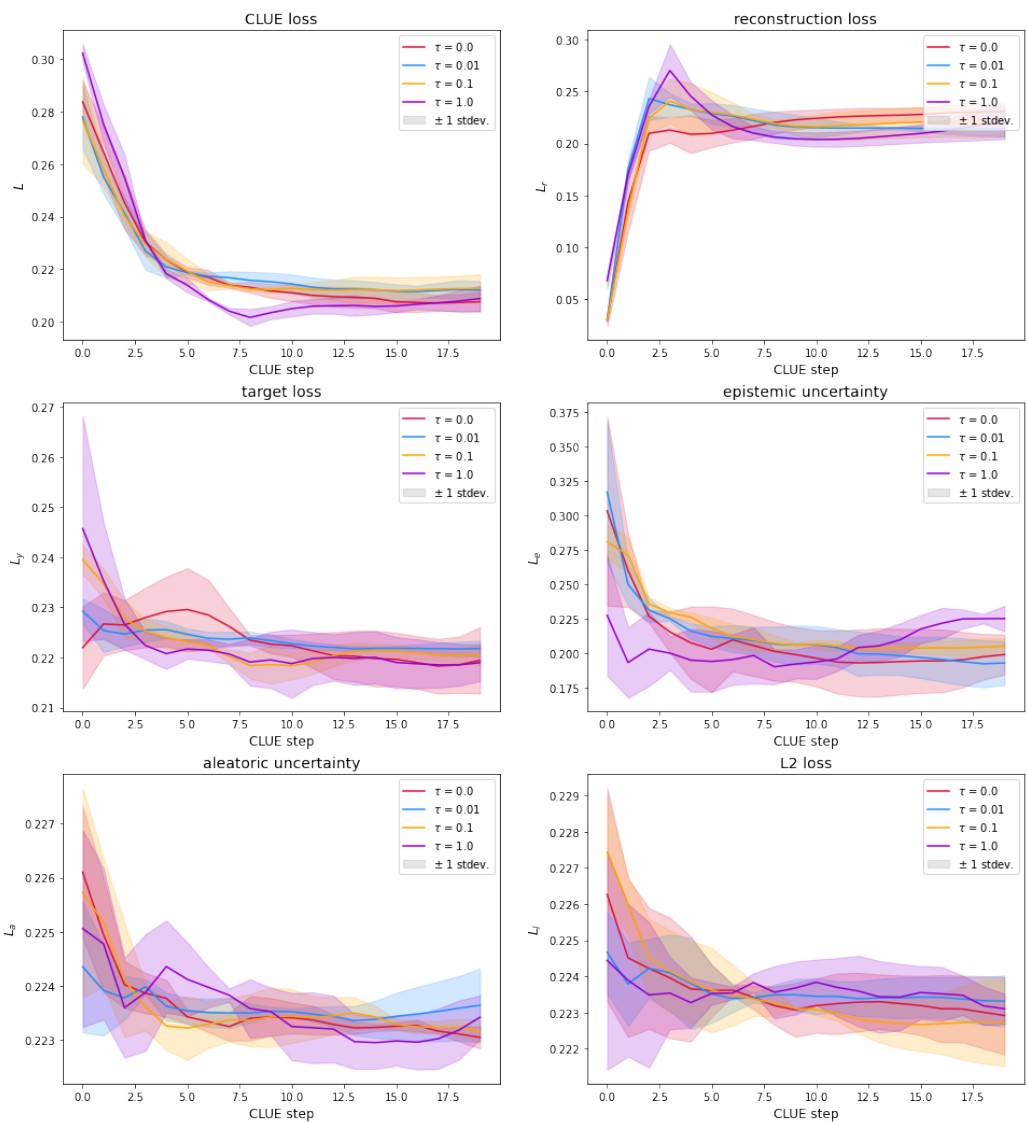

Figure 10: CLUE-optimization loss curves (learning rate $= 1.0$, normalized $\mathcal{L}_{MoleCLUE}$ terms).

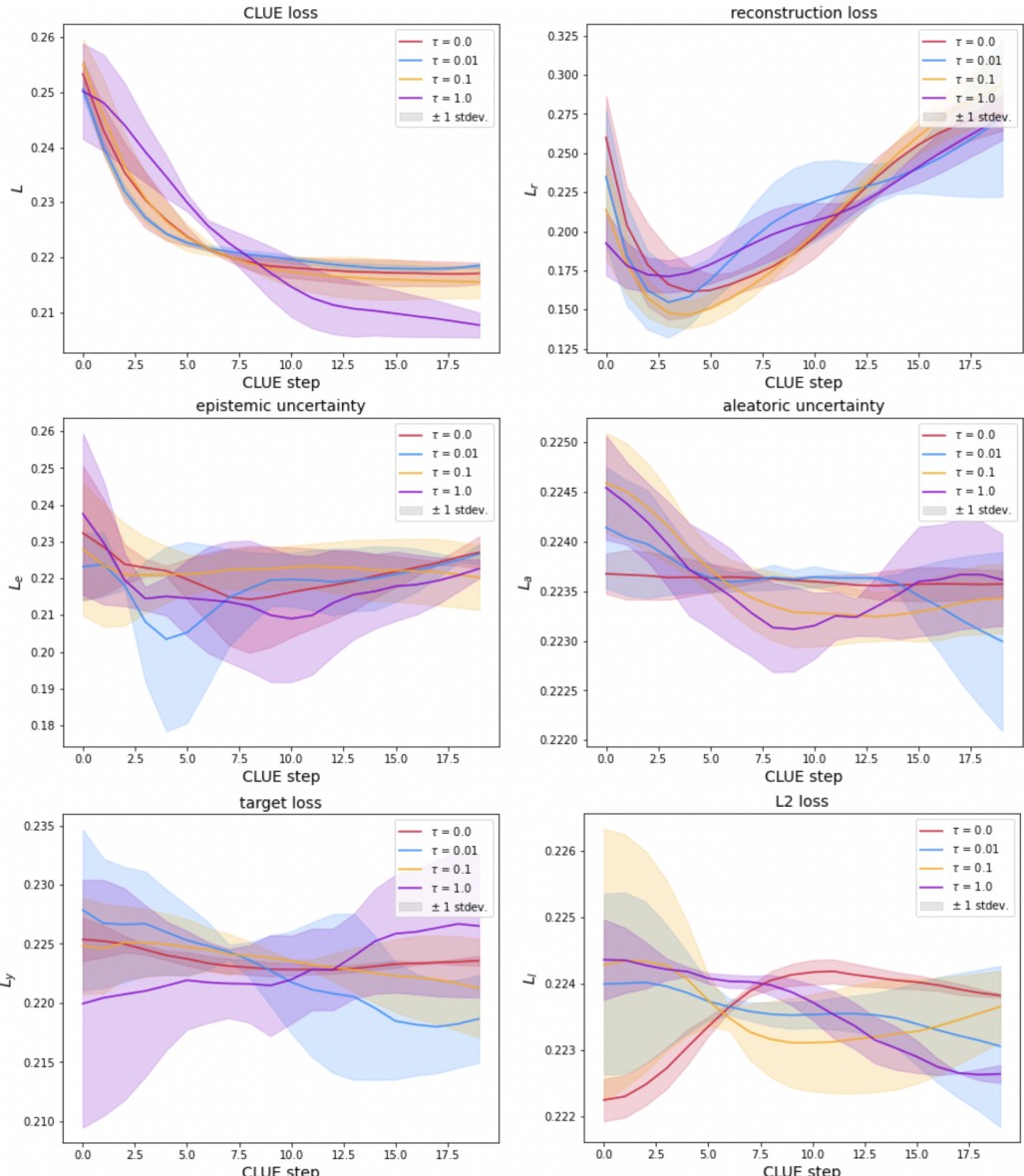

Figure 11: CLUE-optimization loss curves (learning rate $= 0.1$, normalized $\mathcal{L}_{MoleCLUE}$ terms).

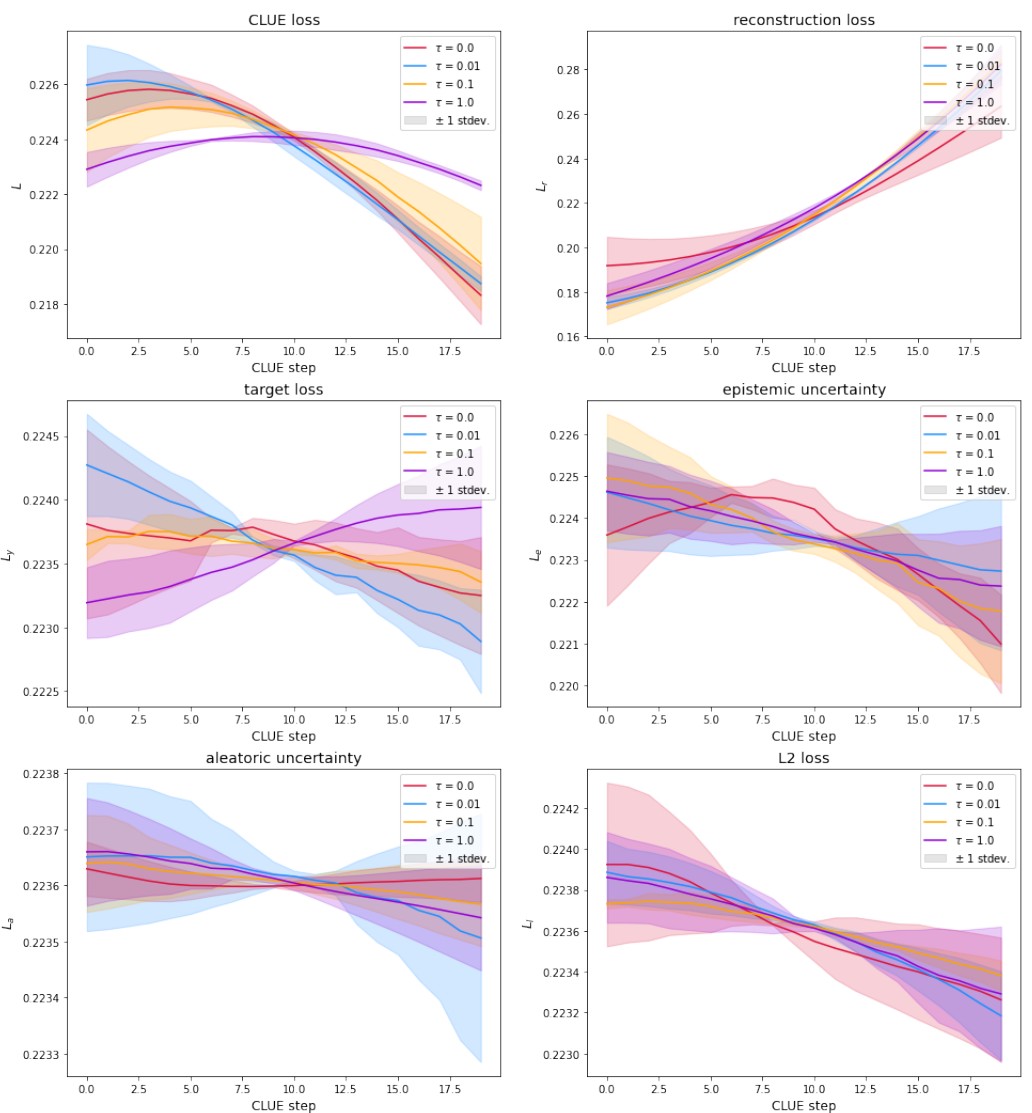

Figure 12: CLUE-optimization loss curves (learning rate $= 0.01$, normalized $\mathcal{L}_{MoleCLUE}$ terms).

## C.2 CLUE trajectories

Following subsection 4.1 and Figure 5, random examples were chosen from the worst-10% of test data points in each CLUE loss term for visual analysis. Trajectories are provided below ordered by term. All examples were optimized with $\tau =$ CLUE LR $= 0.1$ and normalized loss terms.

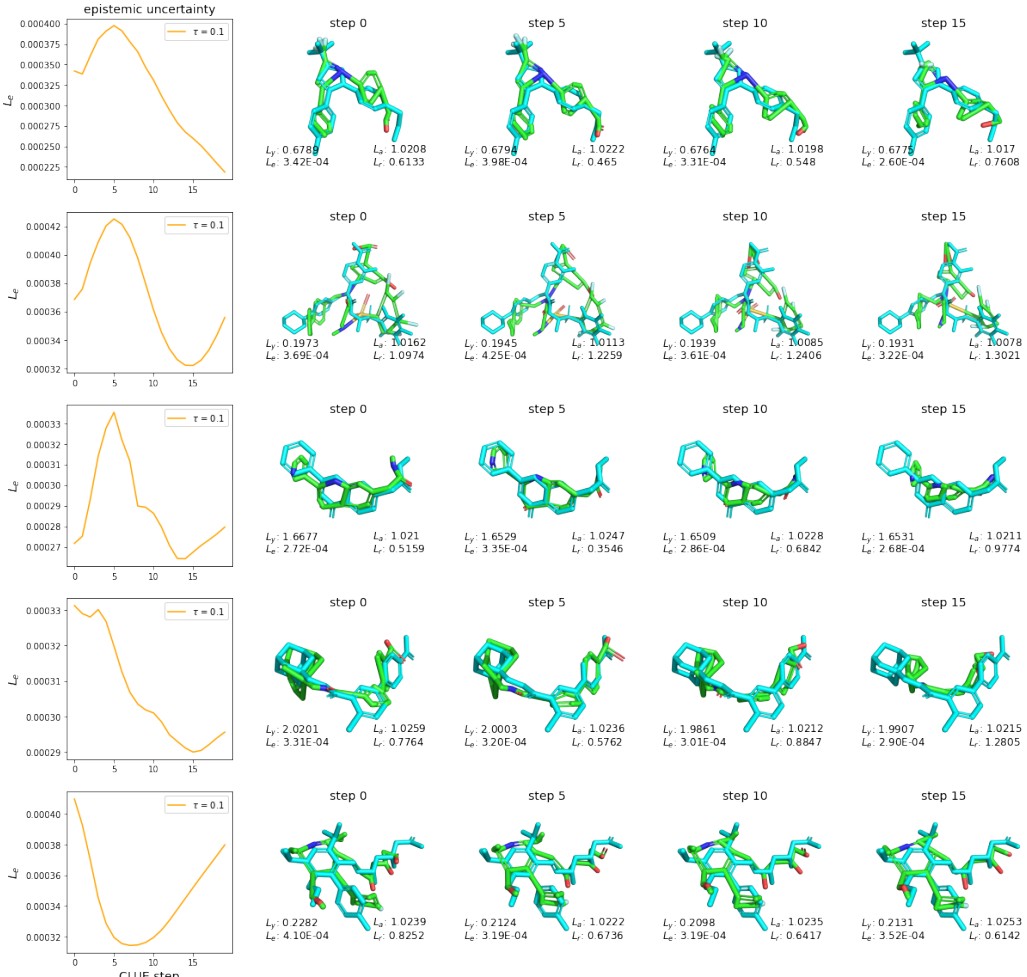

Figure 13: Example CLUE trajectories for worst-10% test data points ranked by epistemic uncertainty.

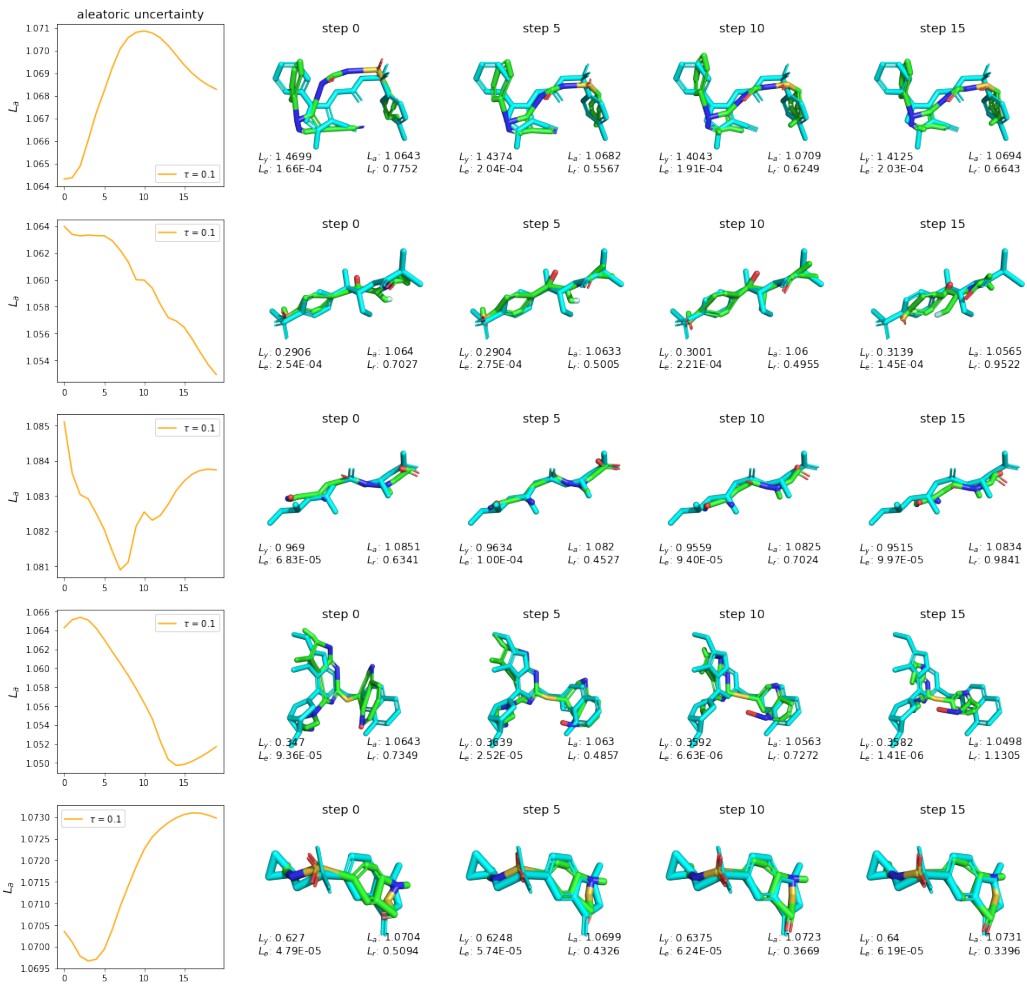

Figure 14: Example CLUE trajectories for worst-10% test data points ranked by aleatoric uncertainty.

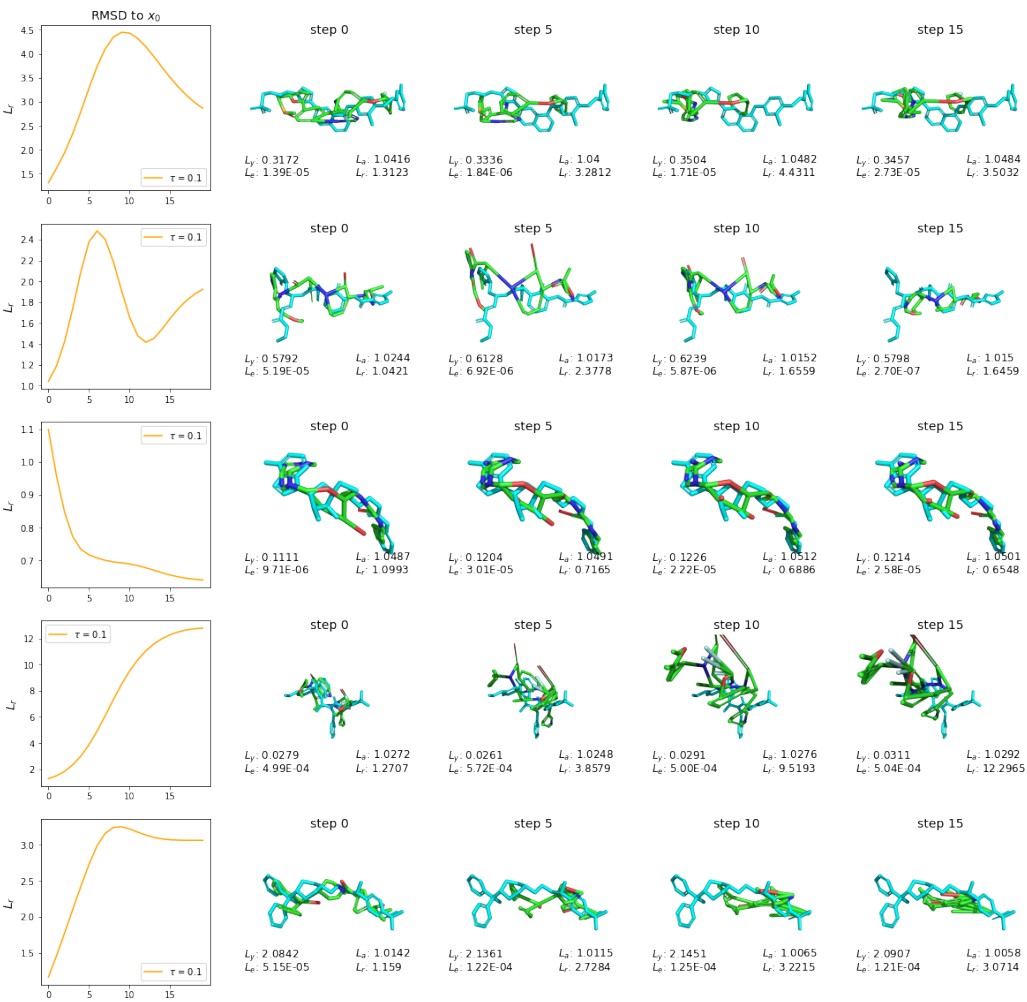

Figure 15: Example CLUE trajectories for worst-10% test data points ranked by RMSD to the input conformer.

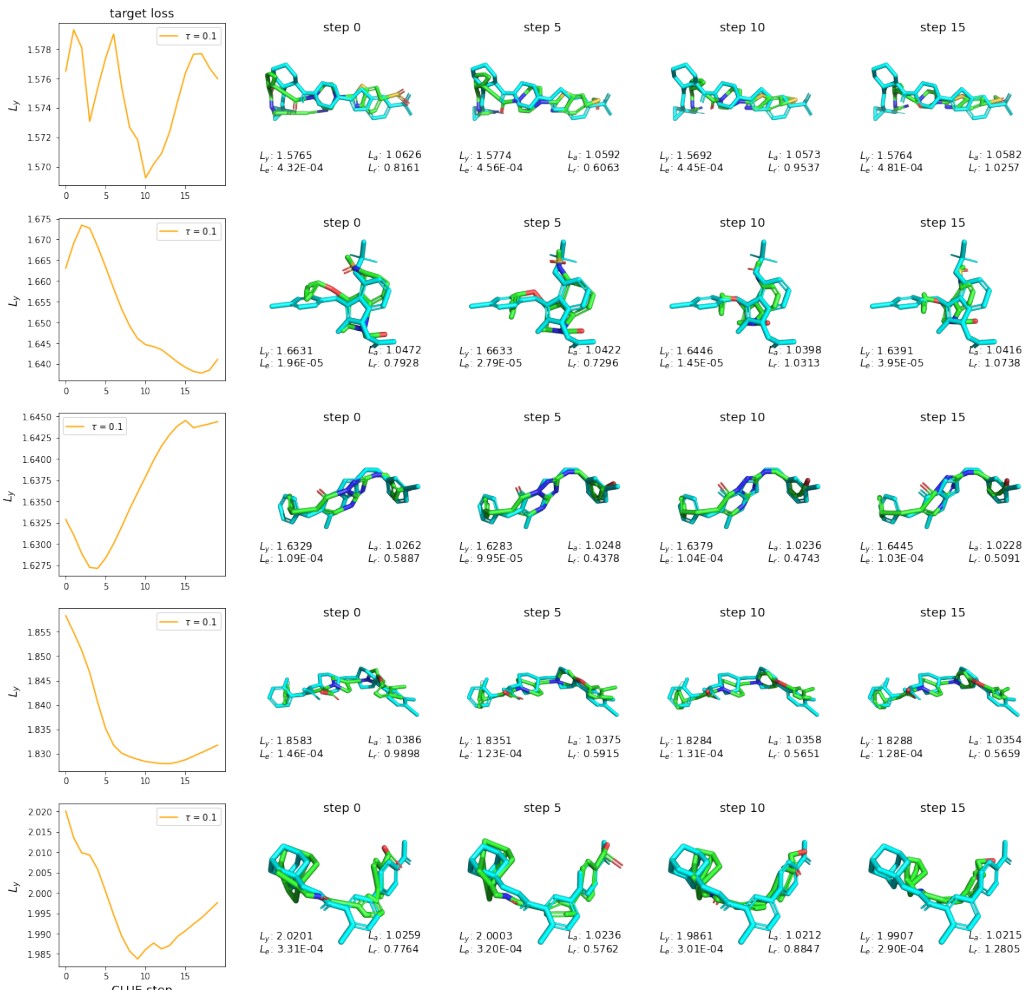

Figure 16: Example CLUE trajectories for worst-10% test data points ranked by target-prediction error.

