# OpenReview forum: "MoleCLUEs: Molecular Conformers Maximally In-Distribution for Predictive Models"
_NeurIPS.cc/2023/Workshop/AI4Science — NeurIPS2023-AI4Science Poster_

### Official Review · Reviewer_Wge3 · 2023-10-19
**MoleCULEs**

**Rating:** 10
**Confidence:** 3

**Review:**

MoleCULEs is presented to generate 3D molecular conformers that explicitly minimize predictive uncertainty. Model performance is assessed in the task of predicting drug-target binding. As the authors pointed out, future directions such as protein property prediction are promising under this work. There could be more assessment tasks and comparative studies, but suitable for now.

---

### Meta-Review · Area_Chair_fFaV · 2023-10-27

**Recommendation:** Accept (Oral)
**Confidence:** 4

**Metareview:**

The paper addresses the challenging task of selecting conformations for models by generating conformers that explicitly minimize predictive uncertainty. It is well-written and convincingly motivated. The quality of the paper surpasses the workshop's standards and, in my opinion, should be accepted.